# Marrying for power: Gendered alliances in mafias

**Maurizio Catino, Alberto Aziani** *, **Sara Rocchi**

Department of Sociology and Social Research, University of Milano-Bicocca, Milan, Italy

* maurizio.catino@unimib.it, alberto.aziani@unimib.it, sara.rocchi@unimib.it

## Abstract

Drawing on judicial records documenting 770 inter-clan alliances through 906 marriages among 623 'Ndrangheta clans, we analyze how matrimonial ties relate to power and cohesion within the organization. Powerful clans occupy central positions in the marriage network, while alliances among less influential clans function as critical "load-bearing" ties—whose simulated removal fragments the network most rapidly. Contrary to a "bride-receiving" narrative, we do not find clear evidence that status alone determines gendered exchange patterns; powerful clans' higher share of incoming brides (0.559 versus 0.437) is not statistically significant. Instead, an inverted-U relationship emerges: clans with moderately high bride-receiving shares (≈55–60%) exhibit the highest centrality, whereas extreme senders or receivers are structurally peripheral. These findings are consistent with marriage operating as an organizational technology—where powerful families build overlapping ties to enhance resilience, and moderate gender asymmetries optimize both lineage consolidation and network brokerage. Overall, the 'Ndrangheta's durability appears to depend as much on peripheral-to-peripheral alliances as on unions of powerful families, underscoring marriage's dual role in maintaining cohesion and managing power.

## 1 Introduction

Since the industrial revolution, individuals in Western societies have become increasingly economically and socially autonomous from their extended families [1–3]. Accordingly, marriages and matrimonial markets are often interpreted as matches between independent individuals who select partners on the basis of love —which has come to be regarded as a prerequisite for matrimonial choice [4,5]— as well as on material and utilitarian considerations [e.g., 6,7].

Nevertheless, even within those societies, there remain "closed" communities, whose membership is based on shared ethnicity, religion, and/or social status and where families continue to play a key role in matrimonial choices [8,9]. In such contexts, marriages tend to be endogamous —occurring within the same ethnic, religious, or social group— and continue to function more as an instrument of rationality than as an expression of romantic love. On the one hand, they are part of

**Data availability statement:** All relevant data are within the paper and its Supporting Information files. In accordance with the PLOS ONE data sharing policy, we have made the matrimonial network publicly available to enhance transparency and replicability. The names of 'Ndrangheta families have been anonymized in order to comply with relevant ethical and legal considerations. In the dataset, powerful clans are identified with the letter 'P'.

**Funding:** The author(s) received no specific funding for this work.

**Competing interests:** The authors have declared that no competing interests exist.

families' strategies aimed at cultural and collective identity preservation [10]; on the other hand, they serve to maintain and expand the family's material and symbolic resources, including status, influence, and material wealth, across generations [11,12]. For instance, a recent study on Romani families [13] highlights that endogamous marriage is a tacit norm. Such marriages are strategically employed both to reproduce the Romani cultural identity and as a mechanism for exchanging wealth and young women between Romani families of comparable social and economic standing.

Families of the modern business elite also continue to function as a "community of interest," and marriage remains a strategically valuable asset for them [14,18]. Even after the rise of the joint-stock company —characterized by separation between ownership and management— these families have remained a crucial factor in the success of their firms. One reason is that familial networks provide social capital, which enhances efficiency by reducing transaction costs [14]. Marriage, by creating new and durable relationships, serves as a primary mechanism through which families accumulate social capital [15]. Through marriage, families gain access to material and symbolic resources, open up new business opportunities, and preserve, or even enhance, their social status [16]. For instance, a study of marriages among business families in Taiwan celebrated between 1973 and 2010 found that such families tend to form matrimonial alliances with others who control complementary resources such as, for example, manufacturing capabilities and financial capital. Notably, the economic value of a "good" marriage is also recognized by the market: when a member of a business family announces marriage to someone from a prominent business or political lineage, the family firm's stock price tends to rise [17].

## 1.1 Marriage as an instrument of organizational rationality in 'Ndrangheta

Some authors argue that marriage is used strategically by mafia organizations [e.g., 18–20] and, indeed, "is always strategic, never the culmination of genuine passion" [21, 125]. In this regard, the 'Ndrangheta represents a distinctive case, due to its unique organizational structure. In the Sicilian Cosa Nostra, for instance, the number of relatives within the same organizational unit (the *famiglia*) is intentionally limited to prevent the formation of alternative, informal centers of power that might prioritize the interests of a minority over those of the *famiglia* as a whole [22]. By contrast, the basic organizational unit of the 'Ndrangheta, the *'ndrina*, is a genuinely patriarchal, hierarchical, blood-based family, in which the sons of "men of honor" (i.e., members) are considered young men of honor from birth, even though the ritual of "baptism" that formalizes their membership takes place later in their lives [23,24].

Because the 'Ndrangheta recruits primarily through kinship, marriages function as a strategic tool employed by blood families to seal alliances with other 'Ndrangheta families. Such alliances aim to consolidate power within the organization, ease conflicts with rival groups, access new business opportunities, and/or expand territorial control [18].

In these matrimonial strategies, women are treated as objects of exchange and "given away" to cement such agreements [18], a logic that reflects the tendency of

mafia organizations to relegate women to passive and subordinate roles [25]. A striking case is that of Giulia Immacolata, a woman born in the 'Ndrangheta family Coluccio. In 2014, at the age of just thirteen, she was forced by her parents to become engaged to Cosimo Commisso, nephew of the powerful boss Vincenzo Macrì, to enable her family to gain access to the Commissos' drug trafficking business [26, 1115]. This instrumental role of women reflects the structure of the 'Ndrangheta, described as "an authoritarian organization [...] with unwritten laws, traditional behavior, and interpersonal relationships explicitly shaped by powerful patriarchal family traditions" [19,39].

Beyond material considerations, endogamous marriages also serve important cultural and security functions. Matrimonial ties within the 'Ndrangheta preserve the organization's values across generations, transmitting cultural norms like loyalty, silence (*omertà*), and honor. Historically, women have been central to this process, safeguarding family reputation and reinforcing the 'Ndrangheta's identity by instilling core values in younger generations [23,27]. In this respect, marriages also contribute to secrecy and, consequently, enhance organizational security. Indeed, the 'Ndrangheta has the lowest number of collaborators with justice among Italian mafias [22]. These data can be partly explained by the fact that collaboration with law enforcement agencies entails severing ties not only with the organization, but also with one's own relatives. The cost of such a decision for a member of the 'Ndrangheta is well illustrated by the case of Emanuele Mancuso, a prominent member of a powerful 'Ndrangheta family. To deter him from cooperating with law enforcement, both his mother and his child's mother used his newborn child as emotional leverage, while his aunt applied psychological pressure by appealing to his attachment to his parents: "Come on, your father and mother are your family... how is your mother doing? She's not well! She knows she no longer has a son, how do you think she feels?" [28].

In sum, matrimonial unions, by creating blood ties, ultimately function as a mechanism for fostering trust and cooperation through personal relationships [29]. This aspect is particularly important in the case of the 'Ndrangheta because, as an illegal organization, its members cannot rely on third parties or formal institutions to regulate transactions and resolve disputes. Nor can they readily resort to communication and violence as formal control mechanisms, since both increase the risk of exposure to law enforcement agencies [24,30].

Marriage, as an organizational mechanism for cultural reproduction, resource acquisition, and security enhancement, is therefore a key feature of the 'Ndrangheta. We argue that these practices deserve more in-depth scholarly attention, as they may help explain the organizational puzzle posed by the 'Ndrangheta. As discussed, 'Ndrangheta is the only mafia organization that adopts a kinship-based recruitment strategy, a strategy that typically creates an organizational dilemma [22]. On the one hand, recruiting through blood ties helps overcome information asymmetries about candidates, ensuring the entry of well-socialized and trustworthy members, thereby enhancing organizational security. On the other hand, this approach does not necessarily guarantee the recruitment of individuals with the skills needed by the organization. As a result, kinship-based recruitment may lead to a lower level of human capital, potentially constraining the organization's capacity to manage sophisticated criminal or business activities.

However, this does not appear to be the case for the 'Ndrangheta, which has managed to reconcile a kinship-based recruitment model with organizational and economic success, and is today often described by law enforcement agencies as one of the most dangerous organized crime groups worldwide [31,32]. Organizationally, the 'Ndrangheta emerged in Calabria around the mid-19th century and has since managed to reproduce its structure across several Italian regions [e.g., 22,20] and abroad [e.g., 33–36]. Economically, according to Italian law enforcement agencies, it exhibits a "persistent business-oriented and entrepreneurial vocation" and an "extraordinary tactical versatility" that have no equivalent among other traditional mafia organizations [10,37].

We explore this apparent paradox by analyzing the organizational logic of marriage in the 'Ndrangheta, with particular attention to the role of women and to differences in marriage strategies between powerful and less powerful families. While previous work highlighted the strategic use of marriage among 'Ndrangheta families, much of the existing discussion remains qualitative and largely reliant on biographical sources [e.g., 38], with large-scale quantitative studies still being rare [e.g., 39]. In this respect, the literature still lacks organization-wide, quantitative evidence linking marriage ties

to actual power and organization's cohesion. Specifically, there is little analysis of whether and how the direction of gender exchange—who "gives" versus who "receives" brides—relates to power and network structure.

We contribute to this body of research by examining matrimonial patterns within the 'Ndrangheta from a social network perspective, an approach that, notwithstanding the methodological challenges [see 40,41], has proven to be fruitful in the study of criminal and mafia organizations [e.g., 41–50].

Accordingly, we test whether reputed powerful families sit at the structural core of the 'Ndrangheta matrimonial network, whether the network would fragment faster if their marriages—or instead peripheral marriages—were removed, and whether a clan's share of incoming brides predicts its power and centrality. These tests examine both status versus structure and gender exchange versus power in a way that the literature has not yet systematically attempted.

As such, this article provides the first large-scale empirical investigation of marriage strategies within the 'Ndrangheta, drawing on an extensive dataset that covers hundreds of families and marriages across years. Our analysis reveals the structural centrality of powerful families within the matrimonial network, demonstrating their critical role in maintaining cohesion and influence. At the same time, the findings highlight the complexity underlying the relationship between family power, network centrality, and gender dynamics, challenging simplistic narratives about bride exchange.

The remainder of the article is organized as follows. Section 2 sets out data and methods: we describe the judicial sources underlying the 623–family marriage network, define our proxies for clan "power" and detail the permutation tests, edge-removal simulations, and sliding-window analyses of gender bias. Section 3 presents the empirical results in three parts—status versus structure, network resilience, and gendered exchange—supplemented by robustness checks. Section 4 discusses the implications of these findings for understanding how the 'Ndrangheta balances lineage consolidation with alliance-building, and flags limitations and future research directions. Finally, Section 5 concludes by summarising the main contributions that follow from viewing marriage as an organizational technology.

## 2 Methods

### 2.1 Sources, data, and proxies

**2.1.1 Marriages and network construction.** The empirical backbone of the study is the *'Ndrangheta families database* first assembled by Catino, Rocchi, and Vittucci Marzetti [39]. We rely on the very same corpus and therefore do not replicate their archival discussion in full detail; instead, we summarise the essentials required for the present analysis.

All information originates from two complementary judicial sources: first, genealogical 'family trees' reconstructed from sealed court documents, and second, roughly 50,000 pages of 40 pre-trial orders, indictments, and investigative reports produced by the Antimafia District Directorates (DDA) of Reggio Calabria and Catanzaro, whose combined authority covers the entire Calabrian region, the area of origin of the 'Ndrangheta. These materials provide biographical attributes for more than 4,500 people for whom investigative bodies have proof or suspicion of affiliation to the 'Ndrangheta together with explicit statements of kinship and marriage ties. Previous work has shown that court-derived evidence, while inevitably incomplete, offers unique windows into large criminal networks available to researchers [40,41,51–54] (methodological limitations are discussed in Section 4.1).

The 40 documents were randomly selected following the procedure described below. We first created a database of all criminal investigations conducted under Article 416-bis against the 'Ndrangheta in Calabria that were mentioned in reports issued between 2007 and 2016 by the DIA (Antimafia Investigative Directorate, the Italian multi-force law enforcement agency specialized in anti-mafia investigations). This process resulted in a list of 192 investigations, of which 129 (approximately two-thirds) were led by the DDA of Reggio Calabria and 63 (approximately one-third) by the DDA of Catanzaro. From this set, we randomly selected 40 investigations (25 led by the DDA of Reggio Calabria and 15 by the DDA of Catanzaro), approximately stratified according to the distribution of investigations between the two DDAs.

After retrieving the associated judicial documents, which were kindly provided by the Milan-based DIA, we carried out data extraction and coding manually. First, we entered personal details and relational information for all individuals

included in reconstructed family trees. Second, we retrieved relational and personal information from the selected judicial documents using ten keywords corresponding to Italian terms for family members. Finally, we conducted consistency checks on individuals, marriages, and families, consolidating duplicate entries for both individuals and families.

More specifically, we extracted two types of information from these sources. First, for each named or alleged mafioso we recorded their patrilineal family name, sex, and unique identifier. Second, we recorded every known matrimonial tie by linking the unique identifiers of brides and grooms. Because the individual-level registry and the marriage-tie registry share the same identifier system, they can be combined directly—without any additional disambiguation or probabilistic matching—to trace links across families.

From the marriage records and surname data, we construct a family-level alliance network by encoding each matrimonial tie as an edge between the bride's family node and the groom's family node, when available. The direction of bride exchange does not affect whether a tie exists for cohesion or prominence analyses. For centrality estimates, fragmentation simulations, and descriptive statistics, we analyse the undirected projection. Collapsing an edge's orientation is appropriate for those tasks because they ask only whether a matrimonial tie exists between two families, not who supplied the bride or the groom.

While the undirected edges capture structural cohesion and centrality, the directional information (bride-receiving versus bride-sending) is retained to quantify gendered exchanges from the bride's family node and the groom's family node. This directional framing reflects the patrilineal transmission of surname and the historical practice of patrilocal residence. It aligns with socio-anthropological models of kinship in which women are conceptualized as agents of alliance-building and flows of social capital between families [55,56].

The choice to model clans as nodes, rather than adopting alternative strategies that would allow the analysis of intra-clan heterogeneity of roles, individual-level brokerage, and micro-dynamics stems from the observation –discussed in Section 1.1– that the 'Ndrangheta clans are patriarchal blood families in which marriages are consistently the outcome of decisions taken at the family level, with which individuals have to comply. Consequently, individual or bipartite network models, while analytically valuable for addressing other research questions [see 47,46], are less well suited to capturing the collective and strategic nature of matrimonial strategies in this specific empirical context.

For the purposes of the network-structural analyses, our focus is the interfamily alliance structure generated by marriages between clans. In line with prior work [see 39], we therefore treat unions occurring between members of the same patrilineal clan as a relevant yet distinct matrimonial strategy that complements—rather than substitutes for—inter-clan alliance-making.

Therefore, we remove the 59 consanguineous self-loops—marriages present in the raw data—as they yield no cross-family ties or brokerage opportunities and would only inflate degree counts without adding meaningful alliance information. Consequently, these intra-clan ties do not influence any downstream metrics. This modelling choice does not imply that intra-clan marriages are sociologically irrelevant: rather, we report and interpret their prevalence separately, while restricting the network topology to inter-clan edges in order to evaluate how marital ties sustain the broader structure of alliances across families.

Then, we analyze the remaining largest weakly connected component of the network, which represents its core structure. In a directed graph, a weakly connected component is a maximal subgraph where all nodes are interconnected when edge direction is ignored [57]. This approach ensures we examine families as part of a cohesive relational system. The largest weakly connected component in our dataset comprises 623 families (nodes) and 906 marriage events, accounting for 71.6% of all families and 90.3% of the 1,003 exogamous marriages in the overall network.

Finally, we collapse multiple marriage events between the same two families into a single simplified edge, yielding an unweighted inter-family alliance network. This representation follows established methods in kinship and marriage network analysis [58,59], and reflects our interest in whether an alliance exists between two families, rather than in its frequency. This unweighted network serves as the primary focus of our analysis and contains 770 unique family alliances (edges).

 

**2.1.2 Proxies for power.** Power in kinship-based criminal organizations such as the 'Ndrangheta is neither directly observable nor reducible to a single structural indicator. We therefore adopt a multidimensional approach, combining externally validated measures of clan status with a battery of complementary network-based centrality metrics. Specifically, we rely on an externally sourced classification (described in detail below) identifying powerful clans based on judicial, journalistic, and investigative assessments of their criminal prominence. We then operationalize clan power structurally by computing centrality measures derived from the marriage network, capturing relational dimensions of influence such as direct connectedness, brokerage potential, reachability, and prestige.

Taken together, these measures allow us to assess the degree of alignment between externally perceived status and internal network prominence. While centrality metrics effectively reflect relational aspects of power, we acknowledge that structural position alone does not exhaustively capture all dimensions of clan power—such as economic wealth, coercive capacity, or political influence. Future research could fruitfully integrate additional indicators to enrich our structural perspective.

**Centrality Measures** To operationalise structural power in the marriage network, we compute four normalised centrality indices, each tapping a distinct facet of prominence (Fig 1). Centrality is a classic proxy for power in network theory, where influence stems from position rather than intrinsic attributes [60–62]. Network topology shapes opportunities for control, brokerage and information access; centrality metrics translate those positional advantages into measurable scores [63,64].

The first metric, *normalised degree centrality*, captures a clan's direct matrimonial ties. A high value signals dense immediate contacts and, hence, rapid access to resources and coordination [61]. Second, *normalised closeness centrality* reflects how quickly a clan can reach all others in the network. Families with high closeness require few intermediaries and can disseminate or gather information efficiently [65]. Third, *normalised betweenness centrality* gauges brokerage potential: it is the share of all shortest undirected paths between every pair of clans that pass through the focal clan.

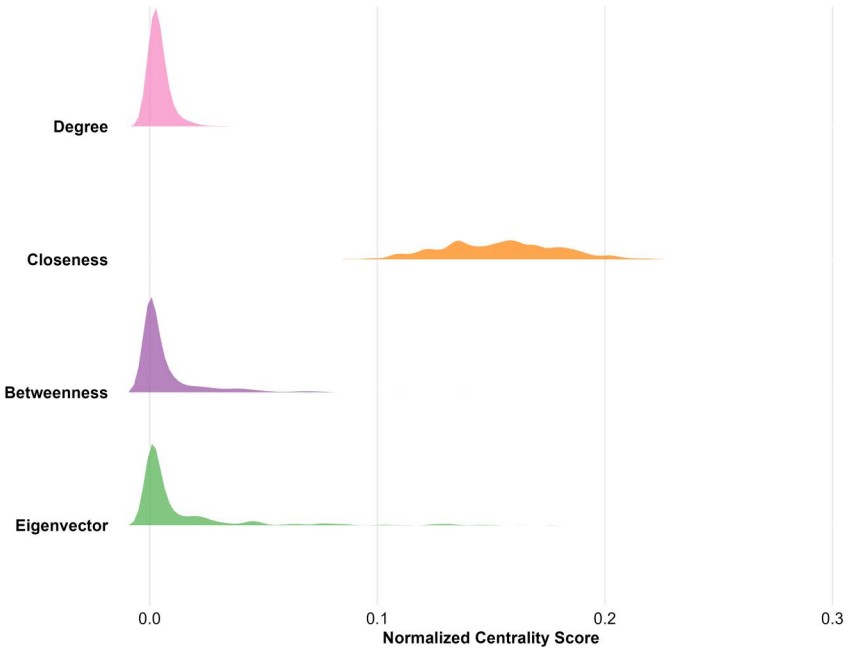

**Fig 1. Distribution of normalized centrality scores in the largest component (*n*=623).** Ridgeline density plots of Degree, closeness, betweenness and eigenvector centralities (textbook-normalized) reveal starkly different shapes: over 50% of clans have zero betweenness and very low degree (medians at 0.002), closeness is more symmetrically distributed (median=0.155), and only one clan achieves the maximum eigenvector score of 1. The horizontal axis is zoomed to [0, 0.3] to focus on the bulk of the distribution.

Under the standard normalisation for undirected graphs, the denominator $(n-1)(n-2)/2$ converts raw counts into a 0–1 scale [66,67]. Finally, *normalised eigenvector centrality* rewards connections to already well-connected families [63]. Let **x** be the principal eigenvector of the adjacency matrix $A$ such that

$$A\mathbf{x} = \lambda_{max}\mathbf{x}. \tag{1}$$

We rescale

$$\widetilde{\text{Eigenvector}}(i) = \frac{x_i}{\max_j x_j} \in [0, 1], \tag{2}$$

so that the most central clan attains 1. This formulation captures the idea that alliances with prestigious partners are associated with additional standing [68].

**Status** To complement network-derived measures of power, we incorporate an external, network-independent ranking of the twenty most influential 'Ndrangheta families identified by the investigative programme *La C Play* [69]. Based on wiretaps, interrogations, video footage, investigative reporting, interviews, and court-anchored inquiries [70], the list reflects qualitative, judicially grounded assessments rather than structural metrics. While institutionally credible, such visibility may also be shaped by media coverage, prosecutorial focus, or clan communication strategies.

We classify every family named in *La C Play* as a "Powerful Clan." Two entries in the source combined surnames—Barbaro–Papalia and Farao–Marincola—which we disaggregate, resulting in 22 distinct clans. Matching these against our network yielded 17 present (Crea, Tegano, Gallace, Nirta, Arena, Giampà, Molè, Morabito, Grande Aracri, Alvaro, Commisso, Barbaro, Papalia, Bellocco, Mancuso, De Stefano, Piromalli) and five absent (Lanzino, Farao, Marincola, Libri, Pesce). The absent clans likely lie beyond the empirical boundaries of our dataset. Families that appear in both sources are coded as "Powerful Clans," while every other observed family is assigned to the "Other Clans" category (Fig 2).

To further validate this classification, we cross-referenced each surname from *La C Play* with the Anti-Mafia Investigation Directorate's (Direzione Investigativa Antimafia, DIA) Semi-Annual Report (I Semester 2023) [37], analyzing the main 'Ndrangheta chapter, regional breakdowns, and annexes detailing investigations and asset seizures. Nearly all surnames appear with varying degrees of prominence, ranging from extensive coverage (e.g., Mancuso, De Stefano, Piromalli, Libri) to more marginal or incidental mentions (e.g., Crea, Giampà, Papalia). Only Lanzino is not cited. This broad alignment reinforces the list's consistency with official threat assessments.

**2.1.3 Bride–groom exchange directionality.** Gender asymmetry in matrimonial strategies is measured using information from the event-level directed marriage graph, in which each individual marriage constitutes a directed tie from the bride-giving clan to the bride-receiving clan. All other analyses employ a simplified undirected counterpart in which repeated intermarriages between the same clans are collapsed into a single alliance.

Specifically, gender asymmetry in alliance building is captured with one normalized statistic per family, the *bride-receiving share*. For clan $i$ let in-degree$_i$ be the number of daughters-in-law (women who marry into the lineage) and out-degree$_i$ the number of daughters the clan gives in marriage. We define

$$\text{bride-receiving share}_i = \frac{\text{in-degree}_i}{\text{in-degree}_i + \text{out-degree}_i}, \tag{3}$$

Lower values of bride-receiving share correspond to families that predominantly send out daughters, while higher values indicate a bias toward receiving brides. The measure is bounded in [0,1]. A value of 0.5 denotes gender-balanced exchange, whereas values above (below) 0.5 indicate a bias toward receiving brides—respectively consistent with consolidating patrilineal capital or forging external ties. This normalisation follows established practice in directional network analysis [64] and is computed on the loop-pruned graph, so consanguineous marriages do not influence the ratio.

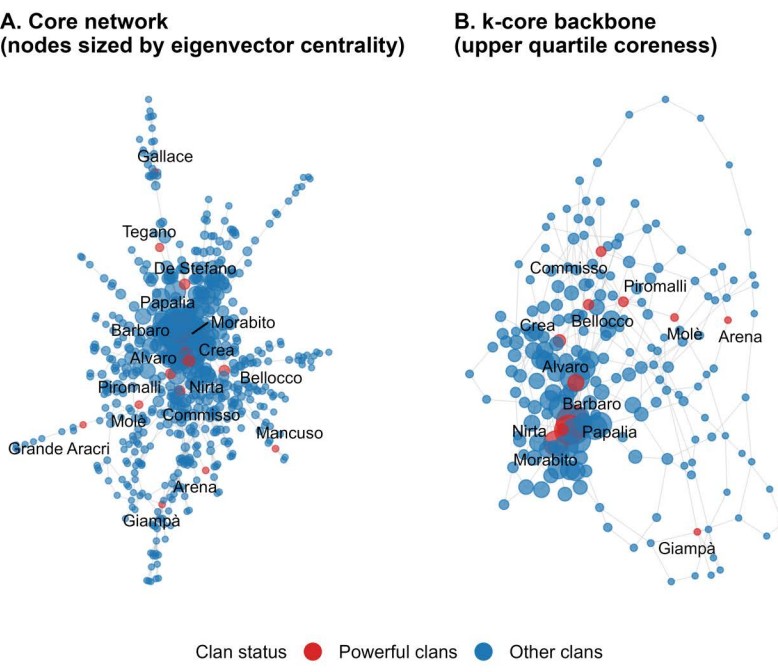

**A. Core network**
**(nodes sized by eigenvector centrality)**

**B. k-core backbone**
**(upper quartile coreness)**

Clan status ● Powerful clans ● Other clans

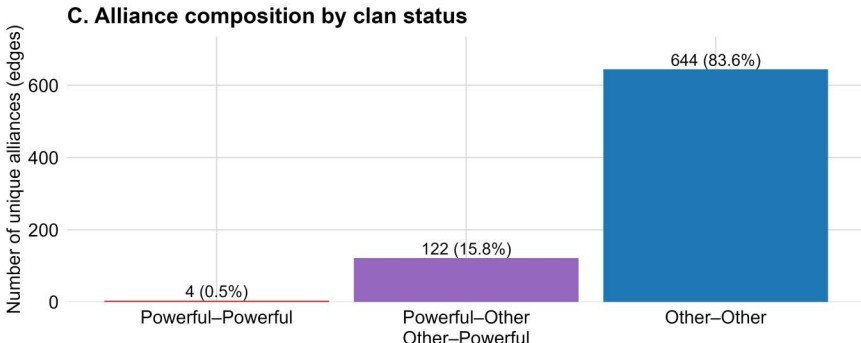

**C. Alliance composition by clan status**

**Fig 2. 'Ndrangheta matrimonial alliance network: structure, core, and tie composition. (A)** Undirected network of inter-clan matrimonial alliances (largest connected component); nodes represent families and node size is proportional to eigenvector centrality; red nodes indicate clans classified as powerful (per the *La C Play* classification), blue nodes other clans; labels show powerful clans. **(B)** k-core backbone of the same network (upper quartile coreness), using the same encoding for node size and clan status. **(C)** Composition of unique alliances by clan status: powerful–powerful (red), powerful–other / other–powerful (purple), and other–other (blue).

The distribution of bride-receiving shares across 'Ndrangheta clans reveals a polarized marital economy with peaks at both extremes of the spectrum (Fig 3). The median is 0.50, while the mean is 0.44, indicating a left-skew driven by a substantial mass of clans with low bride-receiving shares.

## 2.2 Analytical strategy

Our empirical strategy proceeds in four steps. First, we assess whether the clans identified by investigative sources as powerful occupy structurally advantaged positions in the matrimonial alliance network. Second, we examine whether gender-biased patterns of bride exchange are associated with structural influence in the same network.

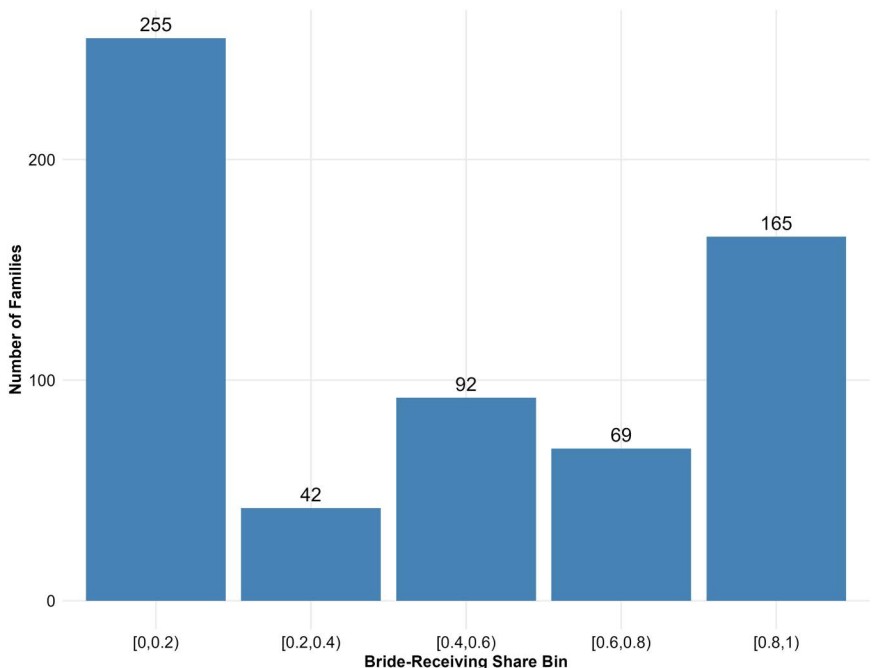

**Fig 3. Distribution of bride-receiving shares across 'Ndrangheta families.** Histogram showing counts of 623 families in the largest weakly connected component. Bin ranges: [0.0, 0.2), [0.2, 0.4), [0.4, 0.6), [0.6, 0.8), [0.8, 1.0]. Bimodal peaks occur at distribution extremes.

First, to determine if clan status is linked to network influence, we analyze our four key centrality metrics—degree, eigenvector, betweenness, and closeness—within the network's largest connected component. We compare these scores between 'powerful' and 'non-powerful' clans using non-parametric permutation tests (subsection 3.1). By randomly reshuffling the 'powerful' labels while keeping the network structure constant, we establish a rigorous benchmark to verify if the observed differences in social prominence are statistically significant or coincidental.

Second, having established whether powerful clans occupy systematically different positions in the alliance structure, we then shift from node-level prominence to edge-level structural contribution. In particular, we assess whether alliances involving powerful clans disproportionately sustain the cohesion of the overall network. If such ties play a central integrative role, removing them should fragment the network more rapidly than removing ties that do not involve powerful clans. To test the network's resilience, we compare the impact of removing edges connected to powerful clans against removing edges between non-powerful clans. We track the resulting loss of cohesion using several complementary indicators (see Appendix A).

Then, third, to study gendered alliance strategies, we compute each clan's *bride-receiving share*, defined as the proportion of its matrimonial ties in which it occupies the bride-receiving role. This measure captures whether a clan tends, on balance, to receive brides into its lineage or to send brides outward through marriage ties. We begin by testing whether clans considered to be powerful display systematically different bride-receiving patterns than other clans (subsection 3.3). Given the modest sample size and non-normal distribution of the data, we opted for non-parametric permutation tests instead of parametric *t*-tests, as the former rely solely on the exchangeability of labels under the null hypothesis. This approach is suitable for network data, where relational dependencies violate standard assumptions of independence and normality, ensuring robust inference without distributional assumptions [60,71–73].

Fourth, we finally turn to the question of how gendered marriage orientation relates to structural prominence in the alliance network (subsection 3.4). Rather than imposing a single global functional form, we estimate a series of local

contrasts along the *bride-receiving share* spectrum, indexed by *c*, comparing clans at each value of *c* to a neutral benchmark centered around parity ($c \approx 0.50$). This approach reveals how network centrality—measured via normalized degree, closeness, betweenness, and eigenvector centrality—varies as clans transition from predominantly bride-sending to predominantly bride-receiving roles.

For example, a positive contrast at $c = 0.60$ indicates that clans with a 60% bride-receiving share are, on average, more central than parity-oriented clans, whereas a negative contrast at $c = 0.90$ indicates lower centrality relative to that benchmark. To ensure that these patterns do not hinge on how narrowly "parity" is defined or on other arbitrary modelling choices, we compute contrasts using both a *strict* neutrality band ($\tau = 0.025$) and a *relaxed* band ($\tau = 0.10$), and replicate the full exercise under a range of alternative specifications and sample restrictions (see Appendix C).

In performing our analyses, for gender-asymmetry indicators such as the bride-receiving share we retain every individual marriage, so the directed graph may contain parallel edges whenever two clans inter-marry more than once. In contrast, all centrality estimates and the resilience simulations operate on a simplified, undirected projection in which at most one edge connects any pair of clans. This distinction ensures that alliance frequency shapes the bride-receiving metric but does not overweight the same clans in the structural-power analyses.

## 3 Results

### 3.1 Status and structural prominence

The clans singled out by journalists and prosecutors as powerful occupy the most advantageous positions in the matrimonial alliance network, as confirmed by 5,000-replicate permutation tests (Table 1).

Powerful families maintain roughly triple the normalized degree of other clans ($\Delta = 0.009$, 95% CI [0.006, 0.010], $p < 0.001$), meaning they are directly connected to 1.2% of other families on average versus 0.4% for non-powerful clans. Powerful clans also exhibit significantly higher closeness centrality ($\Delta = 0.024$, 95% CI [0.012, 0.036], $p < 0.001$), indicating more efficient network access. Likewise, powerful clans command a larger share of shortest paths via betweenness ($\Delta = 0.036$, 95% CI [0.022, 0.043], $p < 0.001$), underscoring their brokerage role, and enjoy a strong prestige advantage in eigenvector centrality ($\Delta = 0.129$, 95% CI [0.075, 0.156], $p < 0.001$). Together, these permutation-based contrasts demonstrate that external reputational authority aligns with internal structural prominence within the marriage network. While these status-based advantages are clear, we further examine the robustness of this relationship across the full range of matrimonial strategies in Appendix C.

### 3.2 Network cohesion and load-bearing ties

While centrality metrics identify powerful families as topologically privileged actors, they do not establish whether these clans' alliances are functionally critical for network cohesion. To evaluate the load-bearing role of powerful family ties,

**Table 1. Network centrality by clan status (5,000 permutation tests).**

| Centrality Measure | Powerful | Other | Difference | 95% CI | *p*-value |
|---|---|---|---|---|---|
| Degree | 0.012 | 0.004 | +0.009 | [0.006, 0.010] | <.001 |
| Closeness | 0.177 | 0.153 | +0.024 | [0.012, 0.036] | <.001 |
| Betweenness | 0.044 | 0.008 | +0.036 | [0.022, 0.043] | <.001 |
| Eigenvector | 0.157 | 0.027 | +0.129 | [0.075, 0.156] | <.001 |

Mean normalized centrality scores (scaled 0–1) are shown for powerful versus other clans. The "Difference" column reports mean differences (Powerful – Other), with 95% confidence intervals computed from permutation distributions. Two-sided permutation *p*-values ($B = 5,000$) test the null hypothesis of no difference. Positive differences indicate structural advantages for powerful clans. Centrality measures: degree (direct connections), closeness (inverse average path length), betweenness (brokerage potential), and eigenvector (recursive influence).

we conducted simulations systematically removing edges under two protocols: proportional and absolute deletion regimes (Fig 4).

Edge betweenness further supports this distinction between prominence and load-bearing importance. While alliances involving powerful clans exhibit higher average edge betweenness (mean=0.0115; median=0.0065) than those among non-powerful clans (mean=0.0082; median=0.0032), the most extreme brokerage ties are not necessarily powerful family-linked (max=0.0804 among non-powerful family alliances). This pattern suggests that powerful families tend to participate in broadly central connections, yet a smaller subset of peripheral marriages can occupy key bridging positions.

Peripheral alliances emerge as critical cohesion anchors. Under equal-proportion removal, eliminating 50% of ordinary marriages reduces reachability by 40%, nearly triple the 15% decline when removing powerful clan ties. This disproportionate fragility persists under equal-count removal: the first 100 ordinary deletions degrade connectivity twice as severely as deletions of powerful-clan ties.

The gradual decay of powerful clan-linked ties signals structural redundancy: powerful clans cultivate multiple, overlapping alliance channels that cushion the impact of losing any one connection. In contrast, the rapid erosion of non-powerful clan ties underscores their role as singular bridges that knit together otherwise distant parts of the network. This divergence highlights how redundancy protects powerful families. Because powerful clans occupy highly central positions and maintain numerous alliances with non-powerful families, they form a structurally redundant backbone of the matrimonial network, even though direct marriages among powerful clans themselves are rare. On the other hand, non-powerful clan unions often span structural bottlenecks; when alliances between less prominent families dissolve, the broader network splits much more quickly.

Largest bicomponent size and average path length within the giant component refine the interpretation of the removal exercise by moving beyond whether clans remain connected to how they remain connected. When edges not involving powerful clans are removed, the network rapidly loses its biconnected core—indicating that cohesion increasingly depends on single "bridge" ties whose removal would fragment the system—and the surviving giant component becomes progressively less efficient, with longer paths between families. By contrast, removing edges involving powerful clans produces a slower contraction of the biconnected backbone and a more contained increase in path length, consistent with a core that is structurally redundant: powerful clans are embedded in overlapping alliance circuits, so the loss of any single powerful clan-linked tie is more easily absorbed without undermining either robust cohesion or internal reachability.

This result has potential operational implications: while powerful clans occupy structurally prominent positions, network integrity and cohesion depend critically on alliances between non-powerful families. Disruption strategies targeting these inter-peripheral alliances would induce faster organizational fragmentation than direct attacks on core families.

## 3.3 Gender directionality and status

Powerful clans receive a greater share of brides (mean=0.559) than other clans (mean=0.437), a difference of +0.122 (Table 2). Although this suggests a tendency for high-status families to act as net receivers, the effect is not statistically significant (95% CI [–0.076, 0.312]; $p$=0.225). The conclusion is unchanged when we additionally control for clans' overall matrimonial activity by permuting status labels within strata of total marriages (degree-stratified permutation; $p$=0.294). Consequently, while the point estimate favors powerful clans, the evidence remains inconclusive.

## 3.4 Gender directionality and network position

Building on this gender-strategy insight, we next examine how variation in bride-receiving share aligns with network position. Across all measures, structural centrality displays a pronounced inverted-U pattern over $c$, peaking between $c \approx 0.55$ and 0.60. This pattern suggests that while parity constitutes a stable baseline, clans that lean modestly toward a bride-receiving orientation occupy the most advantageous positions in the alliance network.

Under the *strict* neutrality definition ($\tau = 0.025$), the strongest and most consistent positive contrasts occur at $c$=0.55 for degree, betweenness, and eigenvector centrality (Fig 5). At this point, clans with a modestly bride-receiving orientation

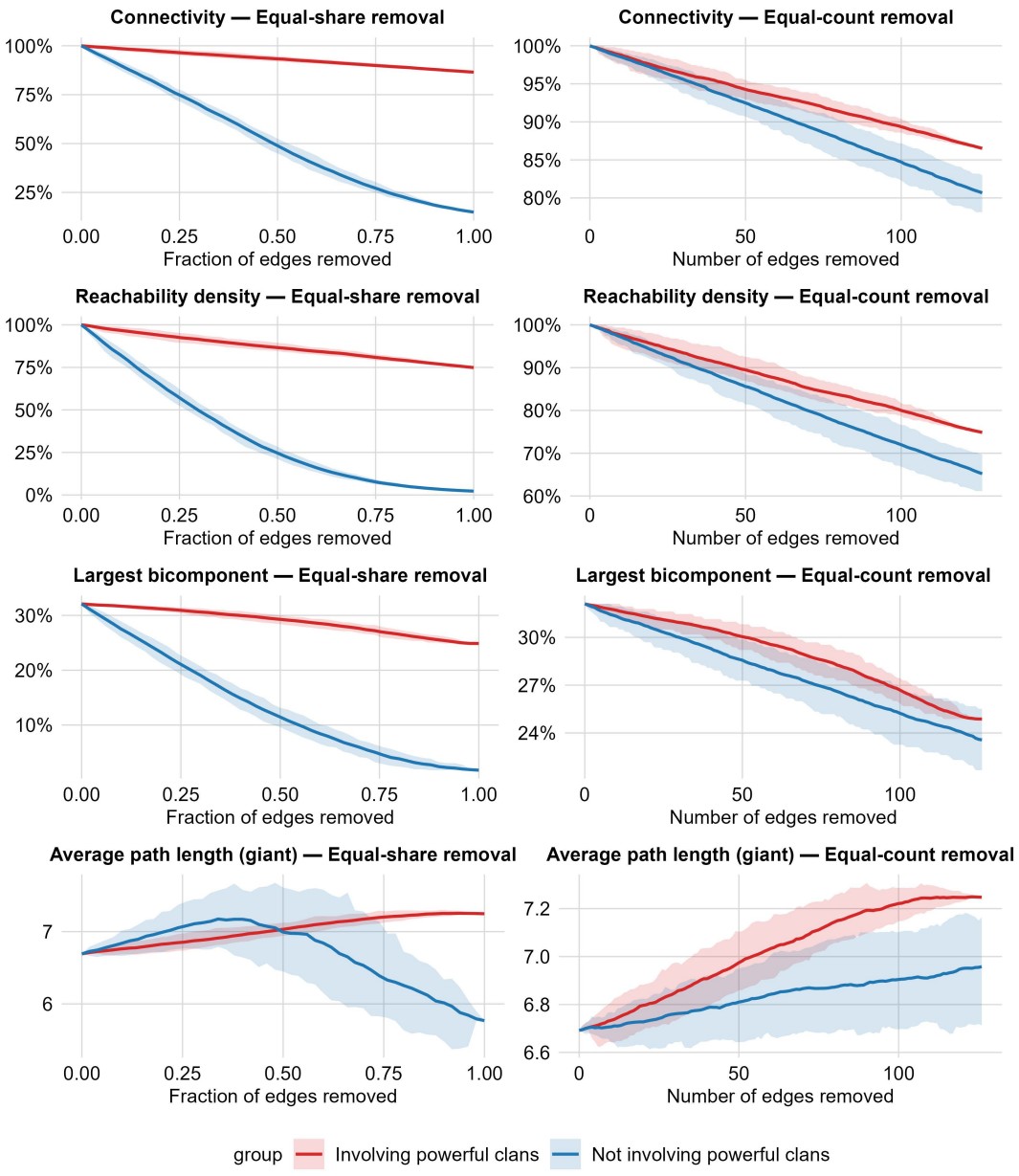

**Fig 4. Resilience of the 'Ndrangheta matrimonial network under two edge-removal schemes.** Monte-Carlo simulations (100 replicates) in which edges are deleted either by *equal-share* (left column) or *equal-count* (right column) schemes, separately for edges *involving powerful clans* (red) and those *not involving powerful clans* (blue). *Rows:* **(A, B)** *Connectivity*—size of the largest connected component as a share of all families; **(C, D)** *Reachability density*—the proportion of unordered family pairs that remain connected by some undirected path; **(E, F)** *Largest bicomponent*—size of the largest biconnected component as a share of all families; **(G, H)** *Average path length (giant)*—average shortest-path length within the largest connected component. *Columns:* (left) *Equal-share removal*: at each step a fixed fraction of that edge category is removed (x-axis = fraction removed within the category); (right) *Equal-count removal*: at each step the same absolute number of deletions is applied to each category (x-axis = number of edges removed, up to the maximum number of powerful-involving ties). Shaded ribbons show the 10th−90th percentile range across replicates.

exhibit higher normalized degree than parity-oriented clans ($\Delta_{\text{degree}} = 0.0092$, $p < 0.001$) and substantially higher betweenness ($\Delta_{\text{betweenness}} = 0.0401$, $p < 0.001$), indicating enhanced brokerage potential. Eigenvector centrality is likewise elevated at $c = 0.55$ ($\Delta_{\text{eigenvector}} = 0.112$, $p = 0.0048$), suggesting that these clans are disproportionately embedded in the network's

**Table 2. Permutation test of bride-receiving share by clan status.**

| Measure | Powerful | Other | Difference | 95% CI | *p*-value |
|---|---|---|---|---|---|
| Bride-receiving share | 0.559 | 0.437 | +0.122 | [−0.076, 0.312] | 0.225 |

Mean shares of incoming brides for powerful (n. = 17) versus other clans (n. = 606), the observed mean difference (Powerful − Other), the 95% effect-centered confidence interval, and two-sided *p*-value based on 5,000 permutations.

most structurally prominent neighborhoods. Closeness centrality follows a compatible, but noisier, pattern: the contrast is positive at $c = 0.55$ but only marginal ($\Delta_{closeness} = 0.0152$, $p = 0.0718$), while the local advantage becomes statistically reliable one step to the right, at $c = 0.60$ ($\Delta_{closeness} = 0.0173$, $p = 0.0108$). Under the *relaxed* neutrality definition ($\tau = 0.10$), results mirror the strict benchmark. The alliance network exhibits its strongest structural advantages among clans that lean modestly toward bride-receiving. The stability of this interior peak across both $\tau$ values suggests that the "sweet-spot" at $c \approx 0.55$–$0.60$ is not driven by an arbitrary definition of parity.

Moving away from this interior interval, advantages attenuate and—especially at the extremes—contrasts turn sharply negative. Under the strict benchmark, heavy bride-senders at $c = 0.00$ are significantly less central than parity-oriented clans across all four measures (e.g., $\Delta_{degree} = -0.0039$, $p < 0.001$; $\Delta_{betweenness} = -0.0137$, $p < 0.001$; $\Delta_{Closeness} = -0.0185$, $p < 0.001$; $\Delta_{eigenvector} = -0.0621$, $p < 0.001$). A comparable structural penalty emerges for heavy bride-receivers at the opposite end of the spectrum (e.g., at $c = 0.90$, $\Delta_{degree} = -0.0034$, $p < 0.001$; $\Delta_{betweenness} = -0.0124$, $p < 0.001$; $\Delta_{closeness} = -0.0161$, $p < 0.001$; $\Delta_{eigenvector} = -0.0625$, $p < 0.001$).

Because bride-receiving shares can be discrete for sparsely connected clans and because local contrasts depend on the definition of the parity benchmark, we re-estimated the full sliding-window analysis under a pre-specified battery of alternative specifications. Across these checks, the central substantive pattern is unchanged: the strongest *positive* contrasts for degree, betweenness, and eigenvector centrality consistently occur around $c \approx 0.55$–$0.60$, and highly asymmetric clans at the extremes remain structurally disadvantaged. These conclusions hold when (i) widening the parity band from ±0.025 to ±0.10, (ii) excluding low-exposure clans (e.g., single-edge families), (iii) tightening window widths and lowering minimum cell sizes, (iv) winsorizing centrality outcomes to dampen outliers, and (v) repeating the analysis with directed prestige outcomes (PageRank and HITS authority) on the directed marriage network. Closeness is comparatively less stable under sparsity—as expected for a path-length measure sensitive to pendant nodes—but the location of the "sweet-spot" remains aligned with the other centrality metrics once low-exposure clans are excluded. Full results are reported in Appendix C (see especially Table 4 and related diagnostics).

## 4 Discussion

Our analysis of the network of 'Ndrangheta interfamily marriages indicates that the most powerful families tend to be those most structurally embedded within the matrimonial network. This finding is consistent with prior scholars, who emphasize marriage within 'Ndrangheta as an organizational instrument of rationality [e.g., 18–21,27,74]. The large-scale analysis we present complements and reinforces existing qualitative evidence that describes how matrimonial ties may serve to facilitate political alliances, business partnerships, and the mitigation of conflicts between rival families. In this sense, our findings are consistent with the idea that marriage is strategically deployed within the organization in ways that are associated with increased influence and expanded territorial or economic reach.

Furthermore, the results of our edge-removal experiments highlight the structural centrality and redundancy of powerful clans, which arises from their extensive connections to a wide range of non-powerful families. This result corroborates the findings of Catino, Rocchi, and Vittucci Marzetti [39] that the network of 'Ndrangheta interfamily marriages has a polycentric structure, characterized by several central families around which denser subnetworks of matrimonial ties develop.

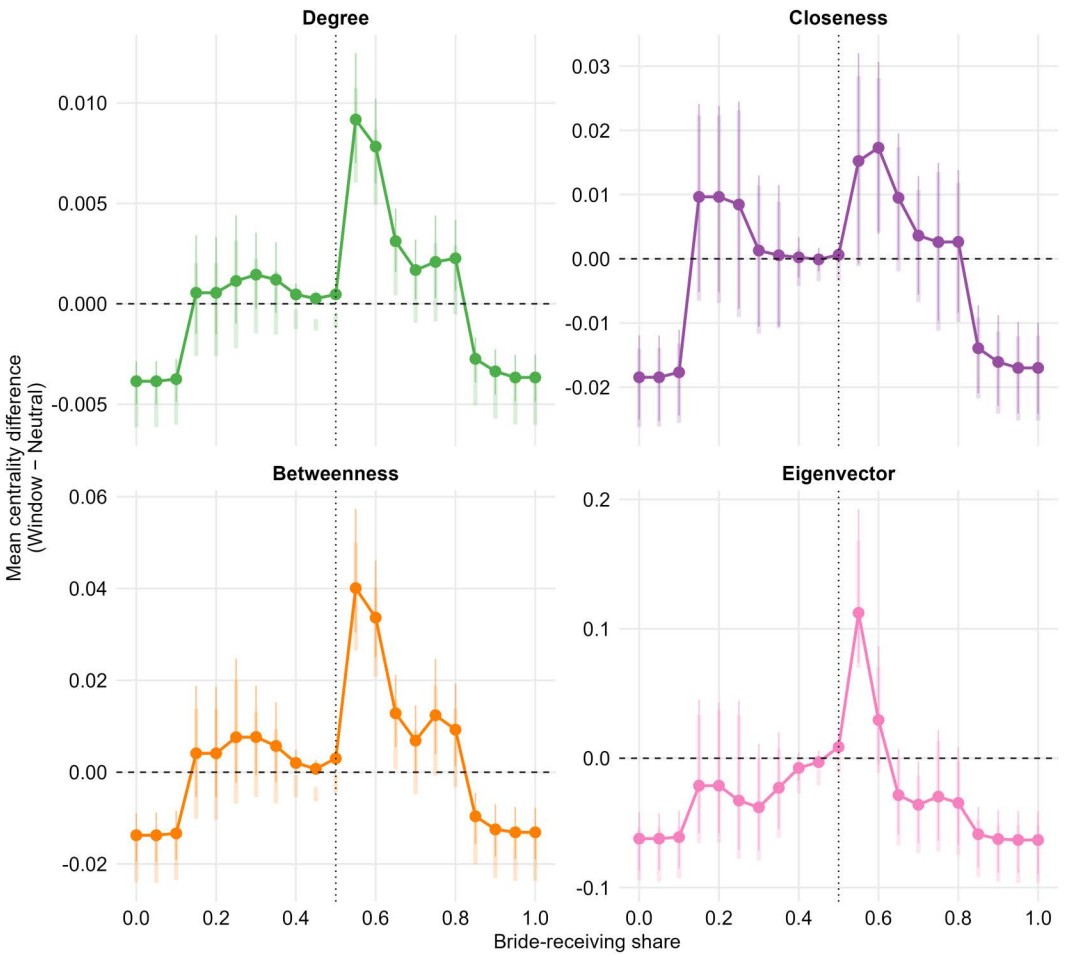

**Fig 5. Adaptive sliding-window contrasts of structural centrality across the bride-receiving spectrum.** Each panel reports local contrasts in normalized centrality as a function of bride-receiving share $c$, where $c = 0$ denotes exclusive bride-sending and $c = 1$ denotes exclusive bride-receiving. The y-axis shows the difference in average centrality between clans near each $c$ and parity-oriented clans ($c \approx 0.50$); values above zero indicate greater centrality than parity, and values below zero indicate a structural penalty. Two specifications assess sensitivity to the parity definition: a *strict* neutrality band ($\tau = 0.025$; thinner/darker intervals) and a *relaxed* band ($\tau = 0.10$; thicker/lighter intervals). Pointwise 95% randomization intervals are obtained from the empirical 2.5% and 97.5% quantiles of $\widehat{\Delta}(c, \tau) - \Delta^{(b)}(c, \tau)$ across $B = 5,000$ outcome permutations; points and lines track the strict specification.

Nevertheless, while reaffirming the centrality of powerful and well-connected families, our analysis also draws attention to the strategic value of marriages among non-powerful families. Our edge-removal experiments show that inter-peripheral alliances are the network's true 'load-bearing' ties—knocking them out fragments cohesion far more rapidly than removing powerful clan-linked marriages. Powerful clans, by contrast, enjoy redundancy in their many overlapping ties, so their own power is buffered against any single rupture. The redundancy observed among powerful clans' alliances might function as a strategic buffer—providing flexibility, protection against targeted disruption, and multiple alternative pathways for resources and influence. Thus, powerful clans' structural advantage lies not in holding fragile critical connections, but rather in strategically cultivated redundancy, ensuring their resilience.

The strategic value of matrimonial-sealed alliances among non-powerful families echoes a dynamic identified by Padgett and Ansell [58] in their study of the Medici network in Renaissance Florence. Padgett and Ansell [58] highlight that the Medici's matrimonial strategy was twofold: on the one hand, they arranged marriages with other noble and powerful

families; on the other, they encouraged intermarriage among those allied families who, because of their lower status (i.e., lower/decayed nobility and wealthy merchants), were deemed unsuitable for direct marital ties with the Medici themselves. While it remains unclear to what extent such marriages among non-powerful families in the 'Ndrangheta are explicitly arranged by the powerful ones, as was the case with the Medici, the strategic importance of these ties suggests that they likely occur with at least the tacit approval of the powerful families. Future research could further explore this dimension of the matrimonial network, particularly the extent to which such alliances are encouraged or even arranged as part of broader power-maintenance/power-enhancing strategies.

Although powerful clans exhibit a higher average bride-receiving share, this difference is not statistically significant—undercutting any simplistic "bride-grabbing" narrative. Indeed, while isolated cases of extreme bride-exchange behavior (such as the documented forced engagement involving Giulia Immacolata) illustrate the potential instrumentalization of women by powerful clans, our comprehensive analysis demonstrates that such extreme examples, although salient, do not represent the broader systematic pattern among powerful families. Instead, most powerful clans adopt more balanced exchange strategies. On the one hand, matrimonial exchanges may reflect a more context-sensitive behavior in which alliance needs, kinship obligations, and local bargaining dynamics all play a role. This does not contradict the notion of women as "precious gifts" used to seal pacts—a hallmark of mafia patriarchy, where women have long embodied the intergenerational transmission of values [19,27]—but it does suggest that power alone cannot predict who gives or receives brides.

Differences in the use of women among the most powerful 'Ndrangheta families may arise from the varying emphasis placed on marriages between and within families [39]. For example, the Barbaro clan regularly marries its daughters into other 'Ndrangheta families to forge alliances, while exhibiting a comparatively lower rate of women from other clans marrying into the Barbaro family. Unlike other families, indeed, the Barbaro displays a tendency toward endogamous marriage, with 25% of unions occurring within the extended Barbaro family –a pattern that appears aimed at preserving internal clan cohesion. On a broader level, the overall tendency among powerful families to adopt more balanced gender patterns—where the number of women sent and received is roughly equal—may also mask underlying differences in family strategies. After all, different types of centrality capture different forms of power, which may in turn guide different matrimonial practices. Some powerful families may aim to maximize the number of matrimonial ties, irrespective of their status (degree centrality); others may prioritize connections with already powerful families (eigenvector centrality); still others may use marriage to enhance their brokerage position within the 'Ndrangheta network (betweenness centrality); and some may pursue a mixed strategy. Such diversity in strategy across powerful families would not be unique to the 'Ndrangheta. Similar patterns have been observed in the Florentine elite during the pre-Medici period, when the significant degree of social mobility produced by marriages was not the result of a unified elite-strategy. Rather, it emerged unintentionally from different familial strategies, each oriented toward maximizing a specific dimension of power—wealth, political influence, or lineage extension. When an elite family arranged a marriage based on one criterion, it inadvertently facilitated mobility with respect to the others, since rarely the three dimensions of power were found in the same family [75].

Our findings nuance existing qualitative accounts. While the instrumental use of women as alliance-builders is well documented historically and ethnographically, our quantitative evidence suggests a more complex picture. Powerful clans do not simply hoard brides; instead, their strategic advantage seems to lie in balancing exchanges, optimizing structural centrality without incurring isolation due to extreme gender bias. This configuration implies participation on both sides of exchange, increasing cross-cutting ties without "locking" the clan into a closed endogamous pattern.

The analysis focuses on the network of inter-family marriages. However, the literature on mafia organizations [18,76,77], together with anecdotal evidence such as the Mancuso case discussed above, indicates that marriage also plays a crucial role within families by fostering internal cohesion, enhancing security, and preventing the fragmentation of assets.

In our network, the intra-family marriages account for 5.56% of all marriages (59 out of 1,062), and are unevenly distributed across clans. On the one hand, they are more prevalent among powerful families (11.17%) than among other clans (4.28%). On the other hand, consistent with Catino, Rocchi, and Vittucci Marzetti [39], we observe substantial heterogeneity even among the most powerful families, suggesting that power asymmetry alone cannot account for variation in the propensity for intra-family marriage. For instance, as discussed earlier, within the Barbaro clan one out of four marriages occurs between family members, whereas the Papalia consistently marries its daughters outside the family, with no recorded intra-family marriages in our network.

These patterns may be partly influenced by the gender composition of offspring within families, which constitutes an exogenous constraint on the range of feasible marital strategies. Nevertheless, the uneven distribution of intra-family marriages suggests that these patterns are unlikely to be driven solely by random demographic variation and may instead reflect family-level choices. In this sense, intra-family marriages are likely the result of an interplay between demographic constraints and clan strategies, rather than from either factor alone.

Future research could further investigate this dimension, as the strategic use of intra-family marriages represents a distinct yet complementary component of clan-level matrimonial strategies, alongside the inter-family unions analyzed in this article.

## 4.1 Limitations

Our findings should be interpreted with consideration of several methodological constraints inherent to studying covert organizations, which highlight opportunities for future refinement.

Data derived from judicial sources prioritizing prosecutable crimes, potentially underrepresent peripheral clans. Such "non-random missingness" is a well-known limitation of court-derived networks and can bias path-based statistics, especially closeness and betweenness, if very central actors were omitted [78]. Specifically, biased geographical coverage may result in the underrepresentation of families operating in some areas, causing them to appear peripheral or absent in the observed network despite potentially playing a more central role in the underlying relational structure. In addition, differences in visibility among clans may inflate the observed centrality of more closely scrutinized actors and contribute to a structure that partly reflects investigative attention rather than underlying relational patterns.

To mitigate these risks, we adopted several safeguards aimed at ensuring sufficiently broad coverage in terms of both cases and groups [79,80]. First, we relied on investigations selected through a stratified sampling strategy across the two Calabrian Antimafia Districts Directorates (i.e., Reggio Calabria and Catanzaro). This approach ensures coverage of the entire Calabrian region –the 'Ndrangheta's historical stronghold– by proportionally representing investigations conducted under both Directorates, thereby reducing the likelihood that central families are omitted for jurisdictional reasons. Second, to limit biases arising from differences in clan visibility, we randomly selected 40 investigations drawn from the full set of the 192 cases reported in the DIA reports over a ten-year period (see the Methodological section). While this strategy does not eliminate the selectivity inherent in judicial data, sampling 40 investigations across a broad temporal window (ten years) helps to reduce the influence of case-specific and time-specific investigative priorities. Third, we cross-checked that information on families identified by journalists and prosecutors as powerful actors is independently confirmed in the DIA's national threat assessments, suggesting that the central core of the network is effectively captured. Fourth, simulation studies show that degree and betweenness-based rankings remain remarkably stable even when many peripheral nodes are missing, and analyses based on arrest warrants or judgments (the very documents we use) reliably identify organizational leaders [e.g., 78, 80]. Finally, because 90.3% of all recorded exogamous marriages fall inside the largest weak component, the observable network retains sufficient density and reach to yield meaningful structural inferences despite inevitable gaps at the margins.

Matrimonial ties constitute only one dimension of relational capital in organized crime networks, alongside mechanisms such as business partnerships, political alliances, and informal patronage relationships [24,39]. While focusing exclusively

on marriage might overlook other influential social ties, the strong correlation between our network centrality measures and independent assessments of clan power (Table 1) indicates that marital alliances indeed capture essential and representative relational dynamics. Furthermore, empirical research on organized crime groups underscores the importance of kinship bonds, particularly marriages, as stable and strategic means of maintaining cohesion, trust, and resource access [21]. Hence, any additional non-marital alliances likely reinforce, rather than contradict, the structural patterns shown by matrimonial networks.

Gender-ratio extremes among marginal clans could reflect sparse data; in these cases, the ratio is discrete, not strategic. To keep such sparse points from dictating the results, we (i) replicated all tests after dropping isolates and single-edge clans, (ii) used sliding windows that automatically widen until they hold at least five—ten in the baseline—observations, and (iii) plotted the pattern with a robust LOESS smoother that down-weights high-leverage cases. These safeguards show that eliminating or winsorizing marginal clans leaves the position and significance of the "sweet-spot" intact; only closeness centrality, which is inherently sensitive to long paths created by sparse nodes, exhibits a modest shift. Residual noise therefore resides on the network's fringe and does not affect the powerful families that drive our substantive conclusions.

We treat matrimonial ties as contemporaneous by collapsing into a single snapshot unions formed over several decades and reported in the judicial documents spanning the years from 2007 to 2016. While this temporal aggregation smooths short-term fluctuations, it potentially inflates estimates of cohesion by implying simultaneous rather than sequential ties. However, consistent with research in other covert networks, relational ties, especially familial ones, typically represent long-lasting commitments that remain significant even if not continuously active [54]. Moreover, our robustness checks via edge-removal simulations (Fig 4) confirm genuine structural fragility rather than artificial hyper-cohesion. Future studies employing dynamic network approaches could clarify how matrimonial ties evolve over time and further validate the sequencing effects of marriage-based alliances.

The *La C Play* designation, while cross-validated with DIA reports, offers a discrete snapshot. Gradations within powerful family tiers exist but would likely strengthen our findings: finer status differentiation might reveal stricter marriage-strategy stratification. That 5/22 clans were absent due to data boundaries highlights the need for multi-source validation—a challenge for all criminal network research.

## 5 Conclusions

Interfamily marriages have long been recognized as a strategic resource in mafia organizations, yet the precise ways in which matrimonial alliance choices relate to power and cohesion within the 'Ndrangheta remained under-explored. By combining judicially reconstructed kinship data with network-analytic tools, we applied three complementary methods: (1) permutation tests to compare the centrality of externally identified "powerful" versus other clans; (2) simulated edge removals to pinpoint which marriages truly hold the network together; and (3) a gendered-exchange analysis of each clan's bride-receiving share to examine how status and marriage-directionality jointly relate to structural influence.

Our results paint a nuanced portrait of matrimonial politics. First, powerful clans indeed occupy structurally privileged seats—boasting higher degree, betweenness, closeness, and eigenvector centrality—validating that reputational authority aligns with positional advantage. Second, the network's cohesion depends disproportionately on marriages among less influential families: these inter-peripheral ties are the true "load-bearing" bridges, while powerful clans are associated with redundant, overlapping unions. Third, although powerful families on average receive more brides, the difference is not statistically significant, dispelling any simple "bride-grabbing" trope. Instead, an inverted-U relationship is observed, where clans with a moderately bride-receiving share (≈ 55–60%) tend to exhibit higher centrality, while those that predominantly send or receive brides are associated with lower structural prominence.

Together, these findings underscore marriage as a calibrated organizational technology—one that both consolidates lineage and extends outreach, at once further enhancing powerful families and binding the periphery into a resilient whole.

## Appendix

### A  Analytical strategy: Formal definitions and robustness checks

**A.1  Status and structural prominence.**  Within the largest undirected component ($n = 623$ families) we compare the four size-corrected centrality indices—degree, eigenvector, betweenness and closeness—between powerful and non-powerful clans. For every measure we generate 5,000 permutation samples that randomly reassign the "powerful" label.

All permutation tests in this study evaluate the null hypothesis that external reputation is unrelated to structural position by randomly reshuffling the *status* label ("Powerful" versus "Other") across clans while keeping the marriage network itself fixed. Under the null, the multivariate vector of centrality scores is *statistically exchangeable* with respect to these labels—every re-labelling is equally likely—so the distribution of any label-contrast statistic is invariant [73, chap. 3]. Because the permutation scheme conditions on the entire observed pattern of kinship and marriage edges, it preserves all within-network dependencies; the only relationship it breaks is the one between status and structural position. The resulting permutation distribution therefore yields an *exact*, finite-sample test of the null, without relying on independence or asymptotic normality assumptions [81].

**A.2  Network cohesion and load-bearing ties.**  We complement the analysis of the relationship between status and structural prominence by investigating whether alliances that involve powerful clans sustain cohesion disproportionately. We perform edge-removal simulations on the undirected marriage alliance graph $G=(V,E)$, where vertices represent clans and an undirected edge indicates at least one intermarriage between members of the two clans. We focus on the largest connected portion of the alliance network and treat parallel marriages between the same pair of clans as a single alliance tie (i.e., $G$ is simplified to remove loops and multi-edges). Clans are labeled as "powerful" based on external media classifications; these labels are treated as exogenous to the marriage network and are used to partition edges into analytically meaningful categories. Each edge $e \in E$ is assigned to $E^{pow}$ if at least one endpoint is a powerful clan and to $E^{non}$ otherwise. Thus, $E = E^{pow} \cup E^{non}$ and $E^{pow} \cap E^{non} = \varnothing$.

At every removal step, we delete edges only—no vertices are pruned—so the vertex set remains fixed; clans whose last alliance tie is removed become isolated nodes. Two complementary deletion regimes are considered. First, an equal-count scheme removes the same number of edges from each subset: at step $k$, we delete $k$ edges sampled from $E^{pow}$ and $k$ edges sampled from $E^{non}$, for $k = 0, 1, \ldots, \min(|E^{pow}|, |E^{non}|)$. This design ensures comparability under an identical absolute number of removals across edge categories. Second, an equal-share scheme removes an identical fraction of edges within each subset: for a fraction $f \in [0, 1]$, we remove $\lfloor f|E^{pow}| \rfloor$ edges from $E^{pow}$ and $\lfloor f|E^{non}| \rfloor$ edges from $E^{non}$, thus standardizing edge removal intensity relative to subset size. In both regimes, at each step, edges are chosen uniformly at random without replacement from the relevant subset, and removals are applied sequentially to generate a perturbed graph $G_k$.

After every removal step, we record multiple complementary cohesion metrics capturing both global connectivity and internal redundancy. First, we measure the relative size of the largest connected component,

$$S_k = \frac{\max_{C \in \mathcal{C}(G_k)} |C|}{|V|},$$
(4)

which captures fragmentation of the alliance system into disconnected blocs. Second, we compute the component concentration index,

$$\rho_k = \frac{\sum_{i=1}^{q} c_i(c_i - 1)}{|V|(|V| - 1)},$$
(5)

where $c_1, \ldots, c_q$ are component sizes in $G_k$; $\rho_k$ is the probability that two randomly selected clans belong to the same connected component, and therefore captures the overall reachability density of the network. Third, to assess cohesion

under the stricter requirement that connectivity remains robust to the removal of any single clan, we track the relative size of the largest biconnected component in $G_k$, i.e., the largest subset of vertices that remains connected after deleting any one vertex. Finally, to capture how efficiently alliances sustain within-component reachability even when the network stays connected, we compute the average shortest path length within the largest connected component of $G_k$. This measure increases as the remaining alliance structure becomes more "stretched" and indirect, even before it fully fragments.

We repeat the entire deletion process 100 independent times for each regime and report the mean curve with 10th–90th-percentile envelopes across replicates. In addition, to examine whether alliances involving powerful clans occupy more bridging positions in the baseline network, we compute normalized edge betweenness centrality for all edges in $G$ and compare its distribution between $E^{pow}$ and $E^{non}$. This provides a complementary, non-simulation-based indication of whether ties attached to powerful clans disproportionately lie on shortest paths connecting otherwise distant parts of the alliance network.

**A.3 Gender directionality and status.** To determine whether politically powerful clans systematically differ in their inflow of brides, we compared the *bride-receiving share*—defined as the proportion of marriages in which a clan assumes the bride-receiving role—between clans classified as powerful and all other clans.

Let $y_i$ denote the bride-receiving share of clan $i$ and let the indicator $g_i$ equal 1 for powerful clans and 0 otherwise. The target estimand is the difference in group means, $\delta = \mathbb{E}(y \mid g = 1) - \mathbb{E}(y \mid g = 0)$, which in the observed data equals $\hat{\delta} = 0.122$. Under the sharp null hypothesis $H_0 : \delta = 0$, the labels $g_i$ are exchangeable; we therefore generated a reference distribution by randomly permuting the power labels across clans while keeping both $y_i$ and the number of powerful clans fixed. Repeating this relabelling procedure $B = 5,000$ times produced the set of permutation statistics $\{\hat{\delta}^{(1)}, \ldots, \hat{\delta}^{(B)}\}$.

The two-sided *p*-value is the proportion of permutation statistics whose absolute value equals or exceeds $|\hat{\delta}|$; in our data $p = 0.225$, meaning that 22.5% of the random reallocations generated a difference at least as extreme as the observed one. This resampling-based *p*-value is exact up to Monte-Carlo error and requires only the exchangeability condition [82].

To quantify uncertainty we inverted the permutation distribution to obtain a 95% effect-centred confidence interval. Specifically, we subtracted the mean of the permuted statistics from each $\hat{\delta}^{(b)}$ to recentre the distribution at zero and then took the 2.5th and 97.5th percentiles, finally adding back the observed estimate $\hat{\delta}$ [see 83].

**A.4 Gender directionality and network position.** Finally, we study how a family's *bride-receiving share* (Section 2.1.3) relates to our four measures of structural prominence—degree, eigenvector, betweenness, and closeness. To this aim, we construct local, distribution-free contrasts that are deliberately robust to sparsity and non-normality in network data. Unlike a regression smoother, this approach estimates a sequence of discrete, local mean contrasts across the bride-receiving spectrum, making non-linearities visible without imposing a global functional form. Unlike the status comparisons in Section A.1, where we permute the powerful-clan label, here we construct a null by permuting the outcome vector (centrality scores) across clans while holding bride-receiving shares fixed.

For each centre $c \in \{0.00, 0.05, \ldots, 1.00\}$, we define an adaptive comparison set

$$\mathcal{W}(c) = \{i : |\mathrm{BR}_i - c| \leq h(c)\}, \tag{6}$$

where the half-width starts at $h(c)=0.05$ and is increased in increments of 0.05 until $|\mathcal{W}(c)| \geq 10$, subject to a maximum half-width of 0.50. This rule preserves fine resolution in dense regions while automatically widening the window—and thus stabilising estimates—where data are scarce.

For the neutral reference group we begin with a parity band around equal exchange and expand it only if needed to ensure stable cell sizes. Formally, for $\tau \in \{0.025, 0.10\}$ we define

$$\mathcal{N}(\tau) = \{i : |\mathrm{BR}_i - 0.50| \leq h_N(\tau)\}, \tag{7}$$

where the neutral half-width starts at $h_N(\tau) = \tau$ and is increased in increments of $\tau$ until $|\mathcal{N}(\tau)| \geq 10$. The strict setting ($\tau = 0.025$) targets clans closest to parity, whereas the relaxed setting ($\tau = 0.10$) increases stability by averaging over a

wider set of near-parity clans; concordant patterns across the two baselines indicate that results are not driven by an arbitrary definition of parity. These clans form the benchmark group against which all local contrasts are computed: at each centre $c$, we compare the mean centrality of clans with bride-receiving share near $c$ to the mean centrality of clans whose exchange is approximately balanced.

For each outcome $Y \in \{\text{degree, eigenvector, betweenness, closeness}\}$ we compute the local mean difference

$$\widehat{\Delta}_Y(c, \tau) = \overline{Y}_{\mathcal{W}(c)} - \overline{Y}_{\mathcal{N}(\tau)}. \tag{8}$$

Positive (negative) values indicate that clans with bride-receiving share near $c$ are, on average, more (less) central than near-parity clans. This permutation test does *not* reshuffle the marriage network. For each $c$, we hold fixed the memberships of $\mathcal{W}(c)$ and $\mathcal{N}(\tau)$ and generate a null distribution by randomly permuting the centrality values $Y_i$ across clans (i.e., permuting the outcome vector). This breaks any association between bride-receiving share and centrality while preserving the marginal distribution of $Y$.

Let $\Delta_Y^{(b)}(c, \tau) = \overline{Y^{(b)}}_{\mathcal{W}(c)} - \overline{Y^{(b)}}_{\mathcal{N}(\tau)}$, where $Y^{(b)}$ denotes the outcome vector after the $b$-th random permutation across clans. Sampling variability is quantified via a two-sided permutation procedure with $B = 5{,}000$ random permutations:

$$p = \frac{1}{B} \sum_{b=1}^{B} \mathbf{1}\left\{ |\Delta_Y^{(b)}(c, \tau)| \geq |\widehat{\Delta}_Y(c, \tau)| \right\}. \tag{9}$$

We report pointwise 95% randomization intervals by adding the $\{0.025, 0.975\}$ quantiles of the permutation (null) distribution to the observed contrast,

$$\left[ \widehat{\Delta}_Y(c, \tau) + q_{0.025}\left(\{\Delta_Y^{(b)}(c, \tau)\}\right), \; \widehat{\Delta}_Y(c, \tau) + q_{0.975}\left(\{\Delta_Y^{(b)}(c, \tau)\}\right) \right],$$

where $q_\alpha(\cdot)$ denotes the empirical $\alpha$-quantile. Inference is distribution-free and does not rely on normality, equal variances, or large-sample approximations. Whenever the 95% pointwise randomization interval lies entirely above zero, we infer that clans with bride-receiving share near $c$ exhibit higher centrality than parity-oriented clans at that location on the spectrum; when the interval lies entirely below zero, they exhibit lower centrality. Substantively, a positive mean difference at $c$ indicates a local structural advantage (e.g., more connections, greater brokerage potential, or more embedded ties to central clans) relative to the near-parity reference group, whereas negative values indicate a structural disadvantage.

Key features of this approach include: (1) adaptive resolution, with window widths expanding only where needed (up to a half-width of 0.50) to stabilise local estimates; (2) local, distribution-free inference via permutation tests induced by permuting $Y$ across clans; (3) retention of comparisons even when $\mathcal{W}(c)$ and $\mathcal{N}(\tau)$ overlap near parity; and (4) diagnostic transparency through exporting the full set of contrasts, confidence intervals, and $p$-values alongside the figure to facilitate replication and sensitivity checks. Scanning $c$ from 0.00 (strong bride-sender) to 1.00 (strong bride-receiver) thus yields a stable, assumption-lean map of structural prominence across gendered alliance strategies.

## B Directed robustness checks: Alliance embeddedness versus exchange hierarchy

Marriage ties can be represented as either undirected or directed relations, depending on the construct of interest. For alliance embeddedness, direction is not substantively relevant: a marriage creates reciprocal, durable inter-family obligations (in-laws, reputational interdependence, and repeated interaction), regardless of whether the bride originates from clan $i$ or clan $j$. Accordingly, all cohesion, centrality, and resilience analyses in the main text use the undirected alliance projection.

However, exchange hierarchy in the marriage market is inherently asymmetric. To capture this aspect, we replicate the status-centrality comparison using a directed clan network in which each tie is oriented from the bride's clan to the

groom's clan (bride family→groom family). This allows us to assess both hierarchical prestige and directed brokerage in marriage exchange. We compute directed centrality measures on the largest weakly connected component and remove consanguineous self-loops.

Table 3 reports 5,000-permutation comparisons between powerful clans and other clans on normalized in- and out-degree, PageRank, HITS hub/authority scores, and directed betweenness. The results confirm that powerful clans remain significantly more central under directed prestige and brokerage measures.

## C  Sensitivity of the bride-receiving share–centrality relationship

The baseline specification keeps *all* clans in the archive, regardless of how many marriages each records, and applies adaptive sliding-window contrasts that begin with a half-width of $h = 0.05$, expand in 0.05 increments, and require at least ten clans on either side of every focal midpoint. This design exploits the entire network while avoiding the volatility that arises from extremely narrow windows or tiny comparison cells.

To verify whether the main findings depend on these choices, we reran the procedure under several alternative settings. First, we widened the reference band around the neutral point $c = 0.50$ from ±2.5 to ±10 percentage points. Second, we excluded pendant clans by retaining only clans with undirected alliance degree ≥ 2 in the collapsed undirected network (i.e., clans connected to at least two distinct partner clans). Third, we combined the degree filter with a narrower sliding-window rule, starting at a half-width of $h(c)=0.02$ and expanding in increments of 0.02 until at least five clans were available per comparison set. Fourth, we winsorized every centrality score at its 95$^{th}$ percentile, damping the influence of extreme outliers. Finally, we applied winsorization and the degree filter simultaneously. Each variant swept the sliding window across the full 0–1 range of bride-receiving shares.

We also verified that the "sweet-spot" pattern does not hinge on computing outcomes on an undirected projection of marriages. For this directed robustness check, we retained the very same sliding-window and permutation design, but replaced the undirected centrality outcomes with two directed measures of prestige computed on the directed marriage graph (bride clan→groom clan): PageRank and HITS Authority.

Table 4 reports, for every specification, the bride-receiving share at which the contrast against the neutral group is largest in absolute value, the corresponding mean difference, and its two-sided permutation *p*-value. Three measures–degree, betweenness and eigenvector prestige–tell a consistent story: their strongest positive contrasts cluster at $c \approx 0.55-0.60$ in most robustness runs and remain significant at the 5 % level or better. Winsorization predictably shrinks the magnitudes, yet the advantages never change sign for $c \approx 0.55-0.60$, confirming that the headline "sweet-spot" is not an artefact of extreme observations. The directed outcomes align with this pattern, with both PageRank and Authority attaining their strongest positive contrasts at $c = 0.55$ ($p < .001$).

**Table 3.  Directed robustness checks of status differences.**

| Directed Measure | Powerful | Other | Difference | 95% CI | *p*-value |
|---|---|---|---|---|---|
| In-degree (norm.) | 0.010 | 0.002 | +0.008 | [0.006, 0.010] | <.001 |
| Out-degree (norm.) | 0.008 | 0.002 | +0.006 | [0.005, 0.008] | <.001 |
| PageRank (rescaled) | 0.393 | 0.137 | +0.256 | [0.203, 0.325] | <.001 |
| HITS authority (rescaled) | 0.058 | 0.006 | +0.052 | [0.044, 0.104] | 0.034 |
| HITS hub (rescaled) | 0.079 | 0.003 | +0.076 | [0.071, 0.132] | <.001 |
| Dir. betweenness (norm.) | 0.005 | 0.001 | +0.004 | [0.004, 0.006] | <.001 |

The directed marriage network encodes bride-family→groom-family. We compute directed centrality measures on the largest weakly connected component (623 clans; 851 directed edges; loops removed), and compare powerful versus other clans using label-permutation tests ($B = 5,000$). "Difference" is the mean contrast (Powerful – Other). PageRank and HITS scores are rescaled by their sample maxima to lie in [0,1].

**Table 4. Peak mean differences in network centrality under alternative specifications.**

| Specification | Measure | Peak $c$ | Peak $\widehat{\Delta}$ | $p$-value |
|---|---|---|---|---|
| Baseline (all clans; $\tau$ = 0.025) | Degree | 0.55 | 0.01 | <.001 |
| Baseline (all clans; $\tau$ = 0.025) | Closeness | 0.00 | −0.02 | <.001 |
| Baseline (all clans; $\tau$ = 0.025) | Betweenness | 0.55 | 0.04 | <.001 |
| Baseline (all clans; $\tau$ = 0.025) | Eigenvector | 0.55 | 0.11 | 0.004 |
| Degree ≥2 ($\tau$ = 0.025) | Degree | 0.55 | 0.01 | <.001 |
| Degree ≥2 ($\tau$ = 0.025) | Closeness | 0.60 | 0.02 | 0.013 |
| Degree ≥2 ($\tau$ = 0.025) | Betweenness | 0.55 | 0.04 | <.001 |
| Degree ≥2 ($\tau$ = 0.025) | Eigenvector | 0.55 | 0.11 | 0.017 |
| Neutral band $\tau$ = 0.10 | Degree | 0.55 | 0.01 | <.001 |
| Neutral band $\tau$ = 0.10 | Closeness | 0.00 | −0.02 | <.001 |
| Neutral band $\tau$ = 0.10 | Betweenness | 0.55 | 0.04 | <.001 |
| Neutral band $\tau$ = 0.10 | Eigenvector | 0.55 | 0.10 | 0.005 |
| Degree ≥2 + narrow windows | Degree | 0.70 | 0.01 | <.001 |
| Degree ≥2 + narrow windows | Closeness | 0.60 | 0.03 | 0.002 |
| Degree ≥2 + narrow windows | Betweenness | 0.70 | 0.05 | <.001 |
| Degree ≥2 + narrow windows | Eigenvector | 0.55 | 0.17 | 0.004 |
| Winsorized (95th pct.) | Degree | 0.60 | 0.01 | <.001 |
| Winsorized (95th pct.) | Closeness | 0.00 | −0.02 | <.001 |
| Winsorized (95th pct.) | Betweenness | 0.60 | 0.02 | <.001 |
| Winsorized (95th pct.) | Eigenvector | 0.95 | −0.02 | <.001 |
| Deg. ≥2 + winsorized (95th pct.) | Degree | 0.60 | 0.01 | <.001 |
| Deg. ≥2 + winsorized (95th pct.) | Closeness | 0.60 | 0.02 | 0.008 |
| Deg. ≥2 + winsorized (95th pct.) | Betweenness | 0.60 | 0.03 | <.001 |
| Deg. ≥2 + winsorized (95th pct.) | Eigenvector | 0.95 | −0.03 | 0.012 |
| Baseline (directed outcomes) | Authority | 0.55 | 0.142 | <.001 |
| Baseline (directed outcomes) | PageRank | 0.55 | 0.224 | <.001 |

$c$ indexes bride-receiving share (0 = extreme bride-sender, 1 = extreme bride-receiver). For each specification and outcome, the table reports the center $c$ at which the absolute sliding-window contrast against the near-parity reference group is largest, the corresponding mean difference $\widehat{\Delta} = \overline{Y}_{\mathcal{W}(c)} − \overline{Y}_{\mathcal{N}(\tau)}$, and a two-sided permutation $p$-value based on $B$ = 5,000 outcome permutations across clans. Specifications are: *Baseline*; *Degree ≥2 only* (single-edge clans excluded); *Neutral ±10 p.p. band*; *Degree ≥2 & narrow window* (initial half-width $h(c)$=0.02, step = 0.02, minimum $n$ = 5); *Winsor 95%* (centralities clipped at the 95th percentile); and the combined *Deg ≥2 & Winsor* variant. Directed outcomes (PageRank and HITS Authority) are computed on the directed marriage graph (bride clan→groom clan) restricted to the same largest weakly connected component, and are rescaled to the unit interval.

Closeness centrality is more volatile than the other metrics, but its overall pattern remains compatible with a non-monotonic, inverted-U relationship. In specifications that retain all clans, the most extreme contrast occurs at the bride-sending end of the spectrum ($c \approx 0.00$) and is negative, reflecting the influence of peripheral clans with long geodesic distances. Across the full range of $c$, however, closeness typically increases toward the interior of the distribution and only declines again at the extremes. This sensitivity arises because closeness averages distances to all nodes and is therefore especially affected by low-degree or pendant clans, whereas Degree, Betweenness, and Eigenvector centrality are driven by more local or recursive structures and display more stable profiles.

Finally, Fig 6 provides a visual robustness check that clarifies the intersection of clan status and matrimonial strategy. While both groups follow a similar distributional logic, the structural advantage of the "sweet spot" is significantly more pronounced for powerful families. For powerful clans (red), the transition toward a balanced-to-moderate bride-receiving

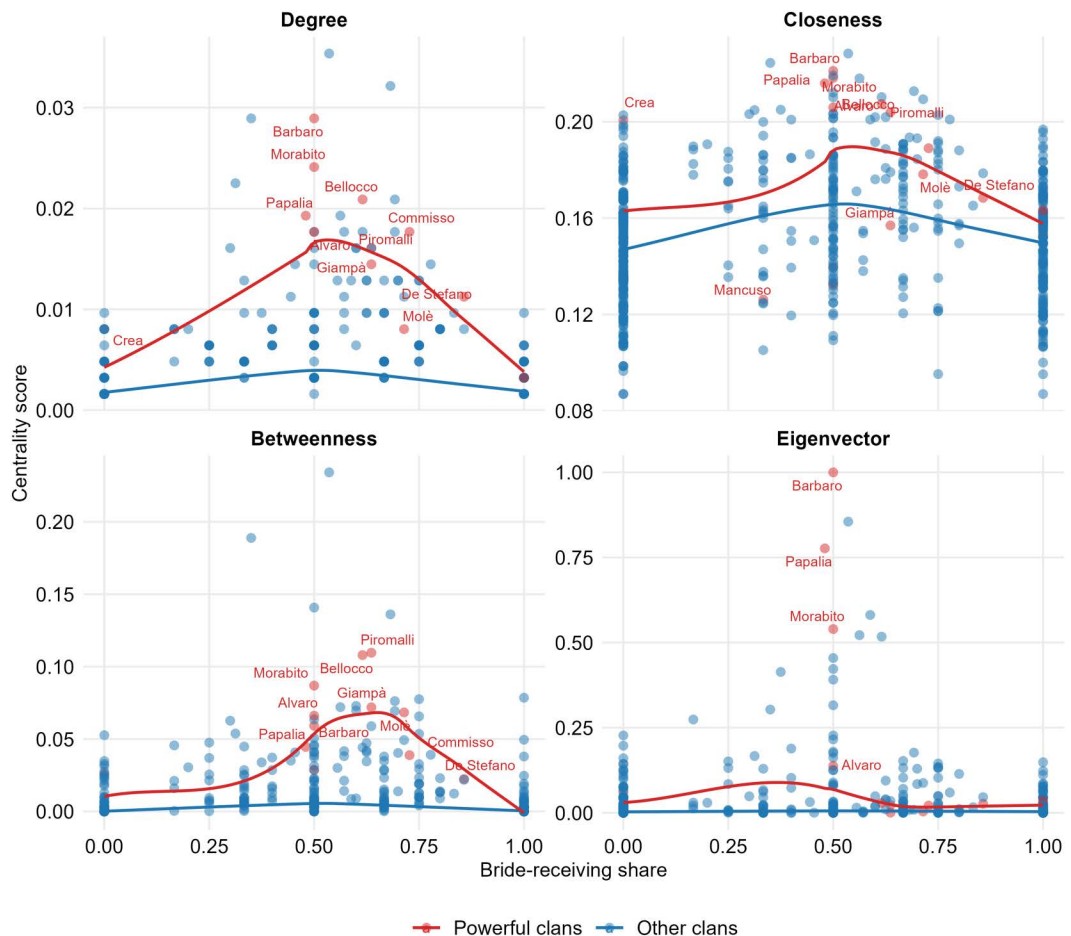

**Fig 6. Status-conditioned returns on matrimonial strategy.** Scatterplots show the relationship between bride-receiving share and normalized centrality, faceted by measure and colored by clan status. Solid curves overlay a *robust, local-linear LOESS* smoother drawn separately for each status group.

share ($c \approx 0.55 - 0.70$) is associated with a sharp, non-linear surge in betweenness and eigenvector centrality—a "prestige payoff" that is far more muted among other clans (blue). This suggests that while the matrimonial economy has a general structural logic, it is the powerful families who most effectively weaponize these positions to consolidate brokerage and recursive influence. Their dominance is not merely a higher baseline of connectivity, but a superior capacity to translate specific matrimonial configurations into disproportionate network power.

## Supporting information

**S1 File. ndrangheta_marriages_adj_matrix_anonymized.**
(CSV)

## Author contributions

**Conceptualization:** Maurizio Catino, Alberto Aziani, Sara Rocchi.

**Data curation:** Sara Rocchi.

**Formal analysis:** Alberto Aziani.

**Methodology:** Alberto Aziani.

**Supervision:** Maurizio Catino.

**Writing – original draft:** Maurizio Catino, Alberto Aziani, Sara Rocchi.

**Writing – review & editing:** Maurizio Catino, Alberto Aziani, Sara Rocchi.

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
