## [Decision Letter · Decision Letter 0]

20 Nov 2025

Dear Dr. Aziani,

Thank you for submitting your manuscript to PLOS ONE. After careful consideration, we feel that it has merit but does not fully meet PLOS ONE’s publication criteria as it currently stands. Therefore, we invite you to submit a revised version of the manuscript that addresses the points raised during the review process.

The manuscript examines matrimonial alliances among ’Ndrangheta families using judicial sources and a clan-level marriage network, highlighting the centrality of powerful clans, the role of peripheral-to-peripheral marriages, and the balance in bride reception. While the topic and dataset are valuable, several methodological issues require clarification: modeling clans as nodes oversimplifies the network and ignores individual-level dynamics; excluding intra-clan marriages removes information on endogamy and internal cohesion; ignoring network directionality limits analysis of gendered exchanges; some figures are difficult to interpret and could be moved or replaced; robustness procedures and the classification of “elite” versus “peripheral” clans need more transparency; potential biases in judicial data should be discussed in greater detail; and the manuscript could be streamlined, references refined, terminology standardized, and data availability described more clearly to improve clarity and reproducibility.

We look forward to receiving your revised manuscript.

Kind regards,

Simon Porcher

Academic Editor

PLOS ONE

Journal Requirements:

https://journals.plos.org/plosone/s/file?id=ba62/PLOSOne_formatting_sample_title_authors_affiliations.pdf....

Reviewers' comments:

Reviewer's Responses to Questions

**Comments to the Author**

1. Is the manuscript technically sound, and do the data support the conclusions?

Reviewer #1: Yes

Reviewer #2: Yes

Reviewer #3: Yes

2. Has the statistical analysis been performed appropriately and rigorously?

Reviewer #1: Yes

Reviewer #2: Yes

Reviewer #3: Yes

3. Have the authors made all data underlying the findings in their manuscript fully available?

Reviewer #1: Yes

Reviewer #2: Yes

Reviewer #3: Yes

4. Is the manuscript presented in an intelligible fashion and written in standard English?

Reviewer #1: Yes

Reviewer #2: Yes

Reviewer #3: Yes

Reviewer #1: This is an exceptionally strong paper. The study analyzes the relation between the marriage networks of 'Ndrangheta families and external status measures of the families. It demonstrates convincingly that high-status families occupy more central positions in marriage networks, while their marriages are less important for the networks’ overall cohesion. The argument is clear, rigorous, and exciting.

In the introduction, you provide an excellent overview of the specific Mafia group under study and why it is a particularly interesting case to study. You embed the work carefully within prior research and clearly describe the contribution of the analysis. What I found particularly convincing is the way you combined different data sources (including a validation of the external power measure). The data set and the construction of the marriage networks are described with exceptional clarity. The rationales for excluding intra-family marriages, focusing on the largest network component, and restricting the analysis to an unweighted network are transparent and convincing. The analytical strategy is rigorous, supported by a wide range of robustness checks and clear justifications for each analytical decision. This thoroughness gave me great confidence in the robustness of the findings.

The results are fascinating. The way external measures of power relate to families’ structural positions in the marriage network is both substantively important and original. The paper is also extremely well-written, making it a pleasure to read. Overall, this is an outstanding contribution, and I recommend acceptance in its current form.

I have only a few minor comments that you may want to consider. I would like to emphasize that these are very minor, and the paper is already excellent and convincing as it stands.

Minor comments

- Figure 2: I was surprised to see that this figure uses a different centrality measure than the four introduced earlier. You might consider using one of the previously introduced measures here for consistency.

- Bride–groom exchange directionality: I was curious to what extent this ratio may be correlated with the (random) gender distribution of children within families, and how much strategic leeway families realistically have. In the discussion, you suggest that some families (like the Papalia family in your example) favor endogamous unions over sending brides to other families. It might strengthen the argument to discuss a bit more explicitly whether this measure is mainly driven by gender ratios or also reflects strategic decisions.

- Figure 4: I found the inclusion of the bride-receiving share on the x-axis somewhat surprising, as this part of the results section is not yet addressing that dimension. Since the only textual reference to this figure (lines 548–550) could also be covered by Table 1, you might consider dropping this figure here, or alternatively explaining its specific purpose and what additional insights it provides.

In sum, this is a fantastic paper, methodologically rigorous, substantively exciting, and exceptionally well-written. It was fun to review this paper. Thanks!

Reviewer #2: This manuscript presents an analysis of the structural impacts of marital ties among ‘Ndrangheta mafia clans. The setting and motivation for the study are interesting and well-described by the authors. The analyses are creative and rigorously detailed; I especially appreciated the authors’ careful attention to finding the right counterfactuals and “null” hypotheses for assessing the significance of observed network patterns. As a whole, the study stands to make a solid contribution to the empirical literature on criminal networks.

Accordingly, I have only very modest recommendations for improving the manuscript, and these recommendations all center on the specific edge-removal tests presented in Figure 5. Here, I actually found the justification for the specific tests carried out to be a little thin. There is not much discussion given to the choice of the two network-structural metrics used to capture network cohesion – largest component (connectivity) and reachability density. To me, these aren’t totally satisfying metrics for giving a strong picture of network cohesion, as most networks contain a large connected component and breaking apart this component is often difficult to do. Have the authors considered also testing for the size of the largest connected bicomponent? And for reachability, what about something like the average path length? And while this would be separate from the edge-removal tests, I also think it would be useful to know whether the edges involving powerful clans score higher than the other edges with regard to “edge betweenness.”

Congratulations to the authors on a very interesting paper, and hopefully these modest suggestions are helpful in further strengthening it.

Reviewer #3: The manuscript examines matrimonial alliances among ’Ndrangheta families using judicial sources and a clan-level marriage network. The authors analyze centrality patterns, gendered exchange dynamics, and network robustness through permutation tests, sliding-window contrasts, and edge-removal simulations. They argue that powerful clans occupy central positions, peripheral-to-peripheral marriages sustain cohesion, and a moderate bride-receiving balance maximizes centrality. The dataset is valuable and the topic important, but several modeling choices, figures, and methodological decisions require clarification.

1) The decision to model clans (surnames) as nodes instead of individuals simplifies the system but removes significant internal heterogeneity, individual-level brokerage, and micro-dynamics. In a context like the 'Ndrangheta, where individuals can play asymmetric roles and intra-clan structures matter, treating each family as a monolithic unit is a strong theoretical assumption. The manuscript should justify this abstraction more fully and acknowledge what information is lost. Alternative modelling approaches (e.g., individual-level networks or bipartite person–clan representations) could produce different insights and should be discussed.

2) The authors eliminate all marriages between members of the same clan because these do not create inter-clan edges. However, intra-family marriages can be substantively meaningful in mafia genealogies, signalling endogamy, closure, or a strategic preference against external alliances. Excluding them entirely may erase a relevant dimension of matrimonial strategy. Even if self-loops are not used in structural metrics, their frequency and distribution should be reported and incorporated into the substantive interpretation. Moreover, rather than being discarded, intra-family marriages could be employed as meaningful weights—either as node-level attributes (indicating family closure) or as tie-strength modifiers—to capture the degree of internal cohesion and reluctance to form external alliances. This alternative would preserve important sociological information without compromising network structure.

3) The dataset naturally yields a directed network: the bride’s family “sends” a woman and the groom’s “receives.” Directionality is analytically important for gendered exchange and hierarchy. Despite this, all centrality measures are computed on an undirected version of the network, and directionality is used solely to compute the bride-receiving share. This inconsistency should be addressed. At minimum, the authors should report whether centrality results hold using directed measures (in-/out-degree, directed betweenness, HITS, PageRank). As it stands, it is unclear whether key findings depend on ignoring direction.

4) Figure 1, showing the entire clan-level network, is visually dense and difficult to interpret. The collapsing of multi-marriages and the clan-level aggregation produce a nearly unreadable graph with limited analytical value. This figure could be moved to supplementary materials or replaced with more informative visualisations (e.g., cluster-focused subgraphs, regional subnetworks, or edge-type distributions).

5) Figure 4, based on adaptive sliding-window contrasts, is also difficult to read. The variable window widths, the logic behind the reference bands, and the meaning of the plotted contrasts are not sufficiently explained. Clearer captions and a more explicit walkthrough of how the figure should be interpreted would be helpful.

6) The two robustness procedures (equal-count and equal-share removal of edges) are potentially insightful, but the rationale and implementation are not entirely transparent. The manuscript should clarify exactly how edges are classified, how many are removed in each scenario, which structural metrics are monitored, and the logic behind comparing these categories. Additionally, because “elite” and “peripheral” clans are defined using external labels, the dependence of the simulation on these categories should be discussed. The current description makes replication challenging.

7) Judicial data are inherently selective and shaped by investigative priorities. While the manuscript acknowledges this, the discussion remains brief. The authors should elaborate on possible biases (geographical coverage, visibility of certain clans, time periods with heavier prosecution) and how these may influence the structure and interpretation of the marriage network.

8) The manuscript is notably long and contains methodological detail that could be moved to Supplementary Materials (particularly the mathematical description of the sliding-window method and some robustness checks). Streamlining would significantly improve readability without sacrificing rigor.

9) Several references appear tangential, outdated, or not directly relevant. A more selective use of literature—especially regarding classical anthropological sources or reports used as scientific citations—would strengthen the theoretical framing and avoid distraction. I suggest to check these works: https://doi.org/10.3390/math10162929 and https://doi.org/10.1109/TIFS.2023.3256706.

10) Terminology for “elite,” “powerful,” and “high-status” clans should be standardized.

11) The Data Availability Statement is adequate, but a brief note on anonymization or preprocessing of judicial material would enhance transparency.

12) A final proofreading pass could remove minor typographical inconsistencies.

.

Reviewer #1: No

Reviewer #2: No

Reviewer #3: No

---

## [Author Response · Author response to Decision Letter 1]

13 Feb 2026

Dear Prof. Porcher, Dear Reviewers,

Thank you for the careful revision and for your appreciation of our work. We have addressed each of the reviewer concerns in detail and implemented the suggested methodological and structural improvements.

Please find the revised manuscript and the attached "Response to Reviewers" file, which contains our comprehensive, point-by-point replies and further explanations of the updates made to the study. We believe these revisions have significantly strengthened the paper and its contribution to the field.

Thank you for your time and for the opportunity to resubmit to PLOS ONE.

Sincerely,

---

## [Decision Letter · Decision Letter 1]

11 Mar 2026

Marrying for Power: Gendered Alliances in Mafias

PONE-D-25-40694R1

Dear Dr. Aziani,

We’re pleased to inform you that your manuscript has been judged scientifically suitable for publication and will be formally accepted for publication once it meets all outstanding technical requirements.

Kind regards,

Simon Porcher

Academic Editor

PLOS One

Additional Editor Comments (optional):

Reviewers' comments:

Reviewer's Responses to Questions

**Comments to the Author**

Reviewer #1: All comments have been addressed

Reviewer #3: All comments have been addressed

2. Is the manuscript technically sound, and do the data support the conclusions?

Reviewer #1: Yes

Reviewer #3: Yes

3. Has the statistical analysis been performed appropriately and rigorously?

Reviewer #1: Yes

Reviewer #3: Yes

4. Have the authors made all data underlying the findings in their manuscript fully available?

Reviewer #1: (No Response)

Reviewer #3: Yes

5. Is the manuscript presented in an intelligible fashion and written in standard English?

Reviewer #1: Yes

Reviewer #3: Yes

Reviewer #1: (No Response)

Reviewer #3: The authors carefully revised their paper according to the Reviewers' suggestions. Therefore, I recommend the publication of their manuscript in its current form.

.

Reviewer #1: No

Reviewer #3: No

---

## [Editor Report · Acceptance letter]

PONE-D-25-40694R1

PLOS One

Dear Dr. Aziani,

I'm pleased to inform you that your manuscript has been deemed suitable for publication in PLOS One. Congratulations! Your manuscript is now being handed over to our production team.

Kind regards,

on behalf of

Pr. Simon Porcher

Academic Editor

PLOS One